# Organization and Community Usage of a Neuron Type Circuitry Knowledge Base of the Hippocampal Formation

**DOI:** 10.3390/biomedicines13102363

**Published:** 2025-09-26

**Authors:** Kasturi Nadella, Diek W. Wheeler, Giorgio A. Ascoli

**Affiliations:** Bioengineering Department and Center for Neural Informatics, Structures, & Plasticity, College of Engineering and Computing, George Mason University, Fairfax, VA 22030, USA

**Keywords:** hippocampus, neuron classification, circuit connectivity, electrophysiology, morphology, spiking neural networks, Google Analytics

## Abstract

**Background/Objectives**: Understanding the diverse neuron types within the hippocampal formation is essential for advancing our knowledge of its fundamental roles in learning and memory. Hippocampome.org serves as a comprehensive, evidence-based knowledge repository that integrates morphological, electrophysiological, and molecular features of neurons across the rodent dentate gyrus, CA3, CA2, CA1, subiculum, and entorhinal cortex. In addition to these core properties, this open access resource includes detailed information on synaptic connectivity, signal propagation, and plasticity, facilitating sophisticated modeling of hippocampal circuits. A distinguishing feature of Hippocampome.org is its emphasis on quantitative, literature-backed data that can help constrain and validate spiking neural network simulations via an interactive web interface. **Methods**: To assess and enhance its utility to the neuroscience community, we integrated Google Analytics (GA) into the platform to monitor user behavior, identify high-impact content, and evaluate geographic reach. **Results**: GA data provided valuable page view metrics, revealing usage trends, frequently accessed neuron properties, and the progressive adoption of new functionalities. **Conclusions**: These insights directly inform iterative development, particularly in the design of a robust Application Programming Interface (API) to support programmatic access. Ultimately, the integration of GA empowers data-driven optimization of this public resource to better serve the global neuroscience community.

## 1. Introduction

Elucidating the neuron type organization of the hippocampal formation is a foundational step to characterize synaptic plasticity in learning and memory [1,2]. While numerous studies have investigated individual properties of the hippocampus [3] and entorhinal cortex [4], integrating this information into a cohesive, accessible framework remains a challenge [5]. To lay the groundwork for this monumental task, Hippocampome.org [6] introduced a framework to systematically define cell types by their properties and to link those properties directly to the underlying peer-reviewed literature evidence [6]. The initial classification system of Hippocampome.org largely relied on the regional and laminar distributions of axons and dendrites as the key determinant to distinguish neuron types [7], thus creating a de facto blueprint for potential synaptic connectivity [8]. Subsequent updates progressively enriched that knowledge base with quantitative neuron type-specific data on molecular transcriptomics [9], firing patterns [10], and in vivo electrophysiology [11]. Further improvements augmented this content by characterizing the directional interactions between neuron types in terms of synaptic physiology [12] and connection probability [13]. Such open access compendium not only enabled researchers to freely and effectively find and explore specific features of interests across hippocampal neurons, but also allowed the extraction of key computational parameters for spiking neural network simulations, including neuron type-specific counts [14] and input-output functions [15], as well as normalized synaptic signals and plasticity [16].

Notwithstanding the promise and general utility of Hippocampome.org for experimental and experimental neuroscientists, it is important to critically assess real-world usage and the engagement of actual users with different data dimensions and interactive features [17]. Explicit citations to the site demonstrate that this resource is actively used by independent scholars for scientific research applications in hundreds of peer-reviewed publications [18]. Nevertheless, these traditional academic metrics do not elucidate how users interact with the web site, which could reveal opportunities for improvement and further development. This study addresses this need by incorporating Google Analytics [19] into the Hippocampome.org portal along with a novel automated service for real-time tracking of user interactions. We present the results of a comprehensive analysis, which revealed research trends and helped guide iterative improvements in accessibility, including the deployment of a robust Application Programming Interface (API) to support programmatic access to Hippocampome.org content. This work thus transcends prior Hippocampome.org publications and updates (v1.0–v.1.12 & v2.0) by adding a new functionality and a quantitative report of user engagement and community adoption of this mature neuroinformatics reource.

## 2. Materials and Methods

User interaction data for the Hippocampome.org knowledge base was collected using Google Analytics 4 (GA4), providing comprehensive metrics such as page views, session counts, and evidence-level interactions. Historical data prior to July 2023 was manually downloaded and archived in structured directories. From July 2023 onwards, automated data collection was implemented via the GA4 API [19], allowing for scheduled retrieval of one day’s worth of data per execution. All GA4 tracking on Hippocampome.org was implemented in compliance with relevant data protection and privacy regulations, including the EU General Data Protection Regulation (GDPR). GA4 does not store users’ IP addresses and only collects anonymized, aggregate usage statistics (e.g., page views, session counts, and geographic regions). No personally identifiable information was recorded or processed. Visitors to the site are presented with a cookie consent banner that enables them to opt in or out of analytics tracking, and usage data are retained only for a maximum of 14 months. As this analysis involved only de-identified, aggregate web traffic metrics, no institutional ethics approval was required.

To facilitate data analysis, we developed a Python-based (v.3) automation pipeline to interface with the GA4 API. This pipeline extracts usage data in CSV format and loads it into a centralized MySQL database. It includes mechanisms to deduplicate previously ingested data, ensuring data integrity, and archives all processed files with timestamps for version control and auditability. The pipeline is configured to run at regular intervals through scheduled tasks using crontab, enabling continuous and up-to-date data collection. To allow and encourage reproducibility, reuse, and further development [20], we released the code open source at https://github.com/Hippocampome-Org/php_v2/tree/GA_analytics_Sept424 (accessed on 10 September 2025). All GA4 usage data is consolidated into general-purpose database tables. A central table in this schema is the page views table, which stores attributes such as page path, view count, timestamp, and relevant metadata. This structure supports scalable and query-efficient access to web analytics data for downstream reporting and analysis.

Rather than storing disaggregated views for each analysis category, we implemented data visualization and analysis at the query level. Custom SQL queries dynamically extracted and reshape data from the general-purpose page views and related tables. This approach enabled the generation of targeted views for functional domain access (e.g., Home, Browse, Tools), monthly and yearly view counts, neuron type-specific page accesses, evidence breakdowns across subregions and attribute types (such as soma locations within the dentate gyrus super-granule layer), and morphology-based feature access (such as axonal lengths).

A lightweight web application was developed using PHP to serve visual analytics dashboards. The interface issues backend queries to the MySQL database and renders results in interactive tables styled after Excel spreadsheets. Each view supports data export in CSV format and dynamic grouping by neuron type, hippocampal subregion, evidence domain, or time period. This interface allows researchers to inspect high-resolution metrics on user interaction with the platform’s data. This code is also released open source at https://github.com/Hippocampome-Org/php_v2/tree/GA_analytics_Sept424 (accessed on 10 September 2025). Geographic usage was visualized using Google Looker Studio [21], incorporating real-time and cumulative access data. Moreover, we plotted monthly page views to analyze temporal trends and engagement changes over time, including pre- and post-2019 comparisons.

We developed and successfully deployed an API to enable flexible, structured querying of the underlying neuron database. This new component and the existing graphical user interface (GUI) share the same PHP-based backend architecture featuring a modular class system for parsing, validating, and executing complex nested queries (e.g., Connection: (Presynaptic: (Name: “Basket”))). We implemented several backend improvements to enhance traceability, performance, and maintainability. We introduced comprehensive query logging with all incoming queries and their associated IP addresses recorded in a dedicated logging table to support auditing and analytical use cases. We applied SQL indexing to critical database columns, leading to a measurable reduction in query execution times, particularly for high-frequency search patterns. Additionally, we refactored the codebase to eliminate redundancy, modularizing the query handling logic into reusable components, thereby streamlining future development and simplifying debugging workflows. For advanced neuron-type searches involving multi-dimensional criteria (such as marker expression, regional projections, and sublayer localization) we extended the system to dynamically generate temporary tables. This strategy significantly improved the speed and consistency of result generation across varied input combinations.

To improve security and access control, we implemented a token-based authentication system for the search API. Users can request a bearer token via a registration form (https://hippocampome.org/php/api/register_token.php (accessed on 10 September 2025)). API access now requires an authorization header (Authorization: bearer YOUR_TOKEN) for all POST requests. All token-based authentication requests are transmitted over secure HTTPS connections using TLS 1.2 or higher, ensuring end-to-end encryption of credentials and payloads. Tokens are stored and validated server-side, with no sensitive information exposed in client-side code or URLs. This update enables user-specific tracking and access logging, supports rate limiting to prevent abuse and ensure fair usage, and restricts exposure of backend endpoints to authorized users. In addition, strict input validation and parameterized queries are now enforced across all endpoints to further safeguard against injection attacks and malformed requests. Public GET-based access is still supported in a restricted form for legacy compatibility.

## 3. Results

Hippocampome.org provides interactive, user-friendly online access to detailed morphological, molecular, electrophysiological, and synaptic properties of all neuron types of the rodent hippocampal formation previously described in the peer-reviewed literature. Here we describe the results of the new automated GA4-based analytics pipeline we designed to capture user interaction data. In order to properly contextualize the analysis in terms of platform functionalities and knowledge dimensions, we first briefly recapitulate the resource organization with illustrative examples and concise summary of neuron type-specific and connectivity features.

### 3.1. Platform Overview: Neuron Types, Properties, and Evidence

Hippocampome.org classifies 180 neuron types primarily based on their main neurotransmitter (glutamate or GABA) and the presence of dendrites and axons through the 26 recognized anatomical subdivisions of the rodent hippocampal formation: dentate gyrus (DG), CA3, CA2, CA1, subiculum, entorhinal cortex (EC), and underlying layers (e.g., stratum oriens, stratum radiatum, etc.). The rationale for this choice is to identify the inherent computational circuit blueprint on the basis of potential connectivity (axonal-dendritic overlaps) and signaling function (excitation or inhibition). On top of this foundational definition, Hippocampome.org extracts and integrates a number of additional qualitative and quantitative features including molecular expression and electrophysiological parameters (Figure 1). In all cases, the resource links each piece of knowledge to the corresponding empirical evidence and related metadata (species and strain, animal sex and age, experimental protocol, and exact literature reference).

Here, the example of vesicular glutamate transporter 3 (VGLUT3) highlights the importance of the complex molecular machinery underlying synaptic dynamics [23] in the determination of neuron type identity.

### 3.2. From Neuron Types to Synaptic Circuitry

Starting from the multidomain characterization of neuron types demonstrated in Figure 1, Hippocampome.org extends to systematically annotate the connection characteristics between directional pairs of neuron types (Figure 2). In particular, unitary synaptic properties are expressed in terms of peak conductance, temporal decay, and short-term plasticity (facilitation/depression). Hippocampome.org normalizes these parameters by best-fitting the classic Tsodyks-Markram model to a large database of experimental data, each carefully annotated with essential metadata (species, sex, age, temperature, recording mode, and dozens additional details). Condition-dependent parameters (e.g., the peak conductance for a particular connection at a body temperature for a female mouse in voltage clamp) are then interpolated using deep learning. Moreover, Hippocampome.org quantifies the connectivity stoichiometry from a pre-synaptic to a post-synaptic neuron type by the connection probability (i.e., what fraction of neuron pairs from those particular types are synaptically connected) and the average number of contacts between connected pairs. These values are derived by computational geometry from experimentally measured axonal and dendritic lengths and layer-specific anatomical volumes. This framework also provides the average distance from the soma of a neuron type-specific connection along the axonal and dendritic trees.

The above illustrations only provide mere vignettes of the publicly available Hippocampome.org content. The resource extends much beyond what static figures can demonstrate not just in terms of the sheer number of neuron types (180) and parametrically characterized synaptic connections (3120), but also in breadth and depth of properties. For example, electrophysiological features also include properties recorded in vivo from live behaving animals, and the expression profiles encompass over 100 molecular markers. In total, Hippocampome.org includes nearly 1300 distinct properties (where “resting voltage” counts as one such property) and 500,000 pieces of knowledge or PoK (e.g., “the resting voltage of DG granule cells is 75 ± 2 mV”). The usage analytics quantified in this report span the full extent of this knowledge base. Thus, Table 1 overviews all of Hippocampome.org properties as organized by data dimension (e.g., molecular markers vs. connectivity), type (e.g., quantitative vs. qualitative), and, within each dimension, numbers of properties and of pieces of knowledge, along with descriptions and examples. Notably, synaptic physiology contributes by far the greatest complexity in the quantification of the rodent hippocampal circuit, underscoring the functional importance of synaptic plasticity.

Hippocampome.org offers distinct functionalities to access this content through an interactive GUI. All properties can be browsed through clickable matrices organized by neuron type and separated by data dimension (e.g., morphology matrix, molecular biomarker matrix, etc.). Moreover, every neuron type can be individually inspected on dedicated pages each listing all properties for that particular neuron type. Furthermore, users can search both properties and neurons through a user-friendly query interface. In all cases, each assignment of a property to a neuron type is systematically linked to the underlying experimental evidence making it immediately available for user inspection. Similarly, synaptic properties and connection probabilities are accessible through matrices organized by pre- and post-synaptic neuron type, with all entries again clickable to retrieve relevant empirical evidence. Lastly, several tools are available to facilitate computational modeling applications, such as a neuronal activity simulator and a workbench to compute synaptic activity, along with an extended documentation to help users navigate through the available content and functionality.

### 3.3. Geographic Reach and Temporal Growth

After this brief recap of Hippocampome.org, we now turn to reporting our detailed analysis of user engagement over the 10 years since the original release leveraging the quantitative results obtained through Google Analytics. All source data for this analysis are available online (https://hippocampome.org/php/analytics_GA4_embed.php (accessed on 10 September 2025)) and continuously updated for dynamic updates. We begin by inspecting the overall geographical distribution of internet traffic to this resource (Figure 3). Visitor access to Hippocampome.org spans the globe, with concentrated engagement in North America and Europe, followed by South and East Asia, reflecting a general trend in neuroscience in general and neuroinformatics in particular [24], but sustained activity also detected in South America, Africa, and Oceania.

Temporal analysis illustrates consistent platform activity and long-term user interest, with cumulative page views rising steadily since 2015 (Figure 4). Usage spikes correlate with substantial content updates such as neuron census integration (v1.11) [14] and the release of synaptic modeling tools (v1.8–2.0) [11,13,14,16,18].

### 3.4. Access by Functionality and Data Domain

Analysis of cumulative worldwide traffic over the entire lifetime of Hippocampome.org reveals the primary modes of user interaction with this resource (Figure 5). When considering overall functionality, more than half of the hits pertain to the Home page, which is expected since it is the primary entry point for the GUI (Figure 5A). Next, it is clear that the most frequent use of the resource comes from the Browse sections, which collectively dwarf access by Search and Neuron Type pages almost 35-fold. The substantive audit of Evidence pages (corresponding to nearly 20% of browse sessions) indicates that a remarkable fraction of users engages in a deep knowledge inspection rather than casual perusal.

Within the Browse category, the Morphology matrix dominates as the most prominent data dimension, accounting for more than half of the hits (Figure 5B). Notably, matrices pertaining to circuitry, including connectivity, synaptic physiology and connection probabilities, account for approximately a quarter of Browse engagement. This highlights the gradual shift in focus from neuron type-centric to synapse-centric between the original classification system in version 1.0 and the simulation-ready framework in version 2.0. Also noteworthy are the engagements with In Vivo Recordings and the Neuron Census, especially considering their relatively recent release (v.1.9 and v.1.11, respectively).

### 3.5. Region, Neuron Type, and Property-Specific Interests

The scope of Hippocampome.org encompasses the whole of the rodent hippocampal formation, but its six constituent subregions (DG, CA3, CA2, CA1, subiculum, and EC) are not uniformly studied in the community. For example, over an order of magnitude more neuron types have been characterized in CA1 than in CA2, and the density of known properties is similarly heterogeneous. It is therefore interesting to investigate whether the patterns of user navigation reflect such uneven distribution of knowledge (Figure 6). A simple analysis by region and neurotransmitter (excitatory/inhibitory) reveals several trends (Figure 6A). First, access to DG neuron types and properties roughly matched the viewed information of all other regions combined. Second, in spite of the well-known greater diversity of GABAergic neurons relative to glutamatergic ones in the cerebral cortex [26], excitatory neuron types are accessed more than inhibitory ones in Hippocampome.org. Third, CA3 has almost as many hits as CA1 despite having about half of its neuron types; and twice the hits of EC despite a similar number of neuron types. More generally, when normalizing the number of hits per region by the total number of (excitatory and inhibitory) neuron types in that region, a clear trend emerges of gradual decrease along the stations of hippocampal processing from DG to EC.

Hippocampome.org internally ranks neuron types by community-derived criteria. Specifically, Rank 1 (R1) only includes one principal cell per region, corresponding to the primary excitatory long-projection neuron (e.g., DG granule, CA3 pyramidal, etc.), and R2 includes the main perisomatic and dendritic-targeting inhibitory interneurons. On the opposite end, R4 includes neuron types with only partial characterization across data dimensions (e.g., unknown molecular expression or unknown electrophysiological properties); and R5 includes neuron types only described in a single publication or with a minimal number (2) of independently observed specimens. Other neuron types not fitting the above criteria are pooled into an intermediate R3. Although this informal ranking is not explicitly disclosed in the Hippocampome.org matrices, we discovered that user access well matches this relative “importance” (Figure 6B). Specifically, the six R1 neuron types attracted more hits than all other neuron types together; and the views normalized by number of neuron types monotonically decrease by rank.

A similar level of heterogeneous user access is reflected not only across data dimensions and neuron types, but also within individual properties and across single instances. For example, not all molecular biomarkers are equally popular within the molecular matrix; and for a given biomarker, users may be more interested in the expression for a specific neuron type than for others. It is impractical to provide here a comprehensive account of the distribution of hits among all properties, although all detailed data are available for further inspection, and dynamically updated, online (hippocampome.org/php/analytics_GA4_embed.php (accessed on 10 September 2025)). Here, we present just a representative “bird’s eye view” of the most frequently accessed properties within six major data domains of Hippocampome.org, including Morphology, Molecular Markers, Membrane Biophysics, Connectivity, Firing Patterns, and In Vivo Recordings (Table 2). For each domain, the most viewed property is listed alongside its total number of views. Additionally, the most viewed specific PoK related to that property is shown with its corresponding view count.

### 3.6. Enhancements to Query System Performance Driven by Analysis of User Data

One of the most surprising findings of our examination of Hippocampome.org usage was the limited employment of the existing Search functionality. We thus decided to investigate possible reasons by inspecting user behavior through the logs of their query patterns. Analysis of these logs revealed frequent use of recurring structured queries targeting specific subsets of neuron types. We hypothesized that slow data retrieval through repeated queries resulted in a frustrating user experience, limiting the adoption of the Search function. At the same time, this suggested an opportunity for optimization through caching strategies. We therefore proceeded to implement backend indexing and dynamic temporary table generation to improve query execution time. Benchmarks comparison of pre- and post-enhancement performance demonstrated a reduction up to 60% in latency for typical multi-parameter neuron-type searches. The faster response may encourage greater utilization of Search going forward. Additionally, the presence of highly complex and occasionally ill-formed query logic in the logs underscored the importance of robust input validation and justified the implementation of robust API access.

### 3.7. Secure Programmatic Access to Hippocampome.org

We refactored the backend query logic into reusable, modular components with standardized query parsing and handling processes. The resulting modules can be used seamlessly by both GUI and API, also leading to significant gains in code maintainability and extensibility. This architectural change allowed for the integration of new filters, such as the neuron types added in v.2.0, with minimal modifications to the existing code base. Furthermore, the modular structure reduced code redundancy, ensuring more efficient debugging and testing workflows and facilitating faster development cycles. The API employs a token-based authentication system. Users register through a dedicated form and receive access tokens, enabling secure, authenticated POST requests. Additionally, we configured infrastructure support for rate limiting and token expiration, providing safeguards against misuse and enabling robust access control over API usage. These enhancements established a secure, controlled, and scalable foundation for API access, effectively replacing the previous unauthenticated GET-based model while maintaining compatibility with select legacy endpoints.

## 4. Discussion

This study provides a comprehensive analysis of user interactions with the Hippocampome.org platform. Tracking detailed usage through GA offered insights into how the neuroscience community engages with web-based neuron type knowledge integrating morphological, synaptic, and molecular evidence.

Our findings show that users primarily access core functionalities such as neuron property browsing and auditing of corresponding experimental evidence, with a significant focus on morphological and synaptic data dimensions [18,27]. These patterns underscore the hippocampal community’s emphasis on anatomy and connectomics [6,14,27]. With respect to neuron types, principal cells and a small subset of inhibitory interneurons in highly studied areas (DG, CA3, CA1) dominate both the experimental literature and user attention. The observation that users frequently access detailed, neuron-specific evidence panels validates the original strategy of Hippocampome.org: to facilitate efficient data exploration and hypothesis generation by linking diverse evidence (e.g., molecular expression, firing patterns, and connection probabilities) to clearly identified neuron types through a set of navigable web pages [6,13,18].

Geographically, Hippocampome.org demonstrates broad international engagement, establishing its value as a global resource. The temporal growth in traffic aligns with major feature updates, confirming that continuous development and expanding content directly enhance community engagement [18,19]. The modular backend design enables further scalability, allowing the addition of future properties (e.g., single-cell transcriptomics) or expansion to other brain regions beyond the hippocampus.

The technical infrastructure introduced in this report, combining a general-purpose analytics database with programmatically defined queries, provides a reproducible and transparent framework for usage tracking. This system not only supports real-time monitoring of research utility but also guides further development by empirical user behavior.

Overall, Hippocampome.org exemplifies a sustainable model for neuroscience knowledge dissemination, combining curation, visualization, and analytics. The emphasis on the rodent hippocampal formation and its circuitry is especially relevant for the study of synaptic plasticity and the neurobiology of learning and memory. Hippocampome.org can be especially useful to support the development of data-driven spiking neural network simulations of oscillatory dynamics [28,29,30] and was instrumental to create large-scale models of cell assembly formation and retrieval [31] and spatial navigation [32].

Future directions include support for simplified computational models (mean-field or neural mass) based on neuron types and cross-species circuit comparisons (e.g., mouse vs. rat), which may facilitate technological applications [33]. Additional opportunities for development also encompass a natural language interface to extract information from the Hippocampome.org knowledge base based on large-language-models [34], integration of citation tracking, and direct export of modeling parameters into simulation environments. As Hippocampome.org continues to expand in scope and depth, the integration of analytics and user feedback will remain central to its evolution as a cornerstone resource in hippocampal circuit research, learning, and memory. As neuroscience continues to scale in complexity and data volume, platforms like Hippocampome.org will play an increasingly central role in organizing and democratizing brain knowledge.

## Figures and Tables

**Figure 1 biomedicines-13-02363-f001:**
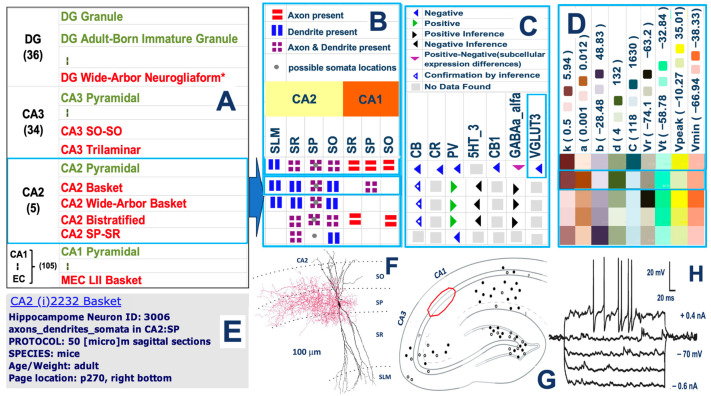
Morphological, molecular, and electrophysiological characterization of hippocampal neuron types. (**A**) Representative excerpt of the Hippocampome.org list of excitatory (green) and inhibitory (red) neuron types across the major hippocampal subregions, with numbers in parenthesis indicating the count of neuron types (e.g., 36 in DG, 34 in CA3, etc.); asterisk indicates types added in v.2.0. (**B**) Laminar distribution of axons and dendrites for the 5 CA2 neuron types across relevant subset of hippocampal layers (SLM: stratum lacunosum-moleculare; SR: stratum radiatum; SP: stratum pyramidale; SO: stratum oriens), with symbol caption above. (**C**) Molecular marker expression for CA2 neuron types, highlighting calcium-binding proteins calbindin (CB), calretinin (CR) and parvalbumin (PV), receptors for serotonin (5HT-3), cannabinoid (CB1), and gamma-amino-butanoic acid (GABAa_α_), and transporters such as vesicular glutamate (VGLUT3). (**D**) Color-coded electrophysiological properties of CA2 neuron types as best-fitted by the 9-parameter Izhikevich model, where k, a, b, and d quantify the dynamic excitability of the neuron, C is the membrane capacitance, and Vr, Vt, Vpeak, and Vmin are the resting voltage, spike threshold, action potential peak, and after-spike reset value, respectively. (**E**) Selected metadata details for a reconstructed CA2 basket interneuron. (**F**) 3D morphological tracing (modified from [22]) of the neuron annotated in panel E, providing experimental evidence for the morphological typing boxed in (**B**). (**G**) Anatomical maps of somatic VGLUT3 expression within a hippocampal slice (modified from [23]). The red ellipse marks the approximate location of area CA2, providing experimental evidence for the molecular typing boxed in (**C**). (**H**) Voltage response to current injections in CA2 basket interneurons (modified from [22]), providing experimental evidence for the electrophysiological typing boxed in (**D**).

**Figure 2 biomedicines-13-02363-f002:**
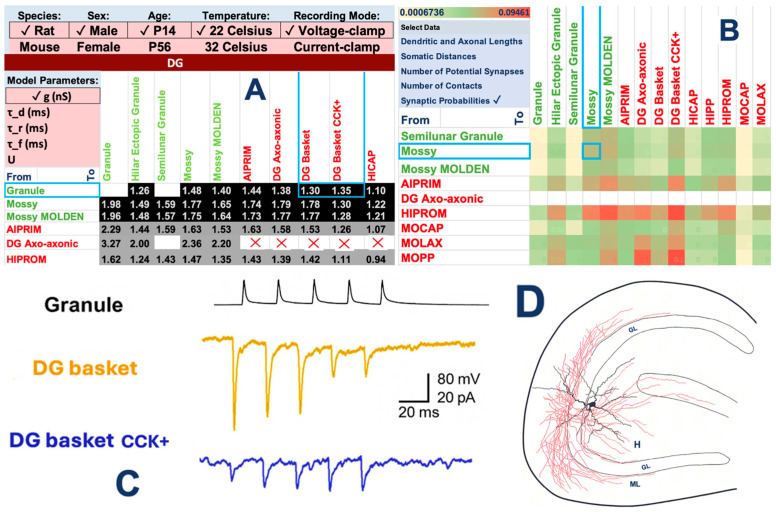
Quantification of neuron type-specific synaptic properties in the hippocampus circuit. (**A**) Normalized synaptic conductance matrix for an illustrative subset of DG neuron type-specific synaptic connections for 2 week-old male rats at room temperature in voltage clamp. Black and gray background respectively indicate excitatory and inhibitory connections, white background denotes inferred absence of connection, and a red X marks an experimentally verified missing connection. The drop-down menus at the top and on the left display respectively other available experimental conditions and other available quantitative parameters, namely temporal decay constant (τ_d) and three short-term plasticity variables: recovery (τ_r) and facilitation (τ_d) time constants, and utilization ratio (U). (**B**) Heatmap of connection probability between an illustrative subset of DG neuron types. The drop-down menu below the warmth scale at the top left indicates the other available variables for each connection. (**C**) Electrophysiological recordings of two DG interneurons in response to granule cell stimulation (modified with permission from [14] *©* 2009 Society for Neuroscience), providing experimental evidence for the values boxed in (**A**). (**D**) Morphological reconstructions of a DG mossy cell with axons (red) and dendrites (black) in the hilus (H), granule layer (GL), and molecular layer (ML), providing experimental evidence for the values boxed in (**B**) (modified from [15]).

**Figure 3 biomedicines-13-02363-f003:**
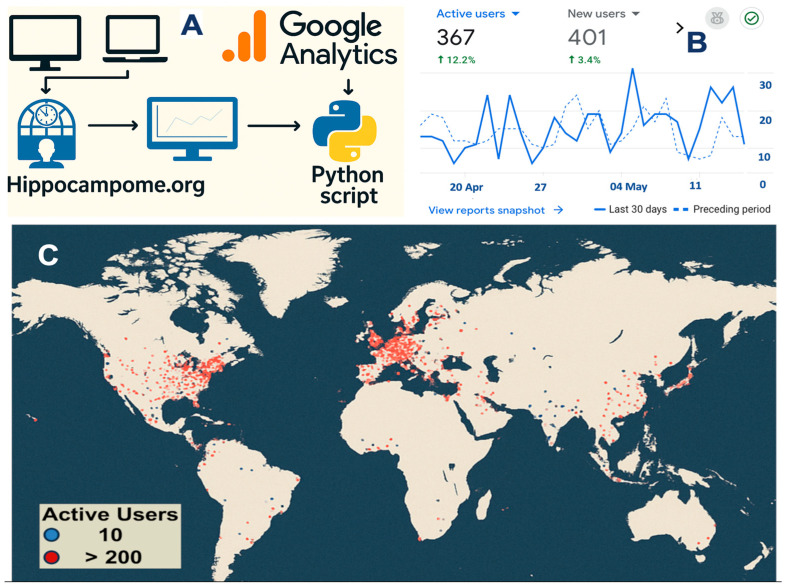
Global distribution of visitors to Hippocampome.org. (**A**) When users access content and functionality on the website from various devices, the resultant interactions are collated via GA and subsequently processed by a Python script. (**B**) Sample GA [25] dashboard with snapshot of key metrics (active/new users) alongside a time-series of user traffic over 30 days (April–May 2025), compared to the preceding period. (**C**) Geographical distribution with color-coded indication of average active users (April–May 2025).

**Figure 4 biomedicines-13-02363-f004:**
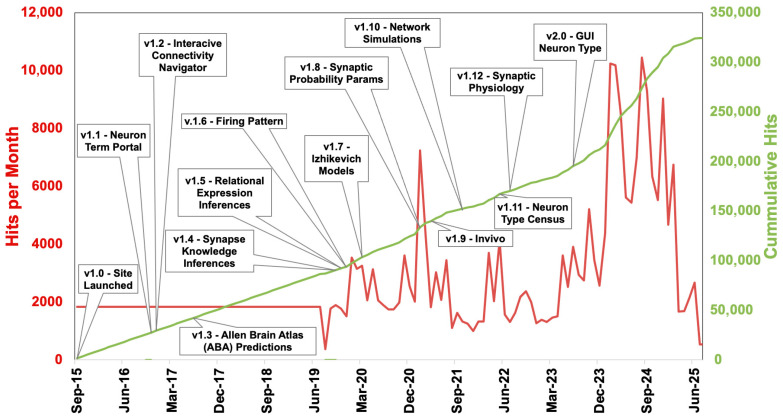
Monthly and cumulative page views over time. User engagement from Hippocampome.org initial launch (September 2015) through the time of writing by 30-day user windows is illustrated by monthly (red/left axis) and cumulative (green/right axis) page views. Key version releases (v1.0 to v2.0) are marked with annotations indicating major feature additions such as the Neuron Term Portal (v1.1), Synapse Knowledge Base (v1.4), and Izhikevich models (v1.7).

**Figure 5 biomedicines-13-02363-f005:**
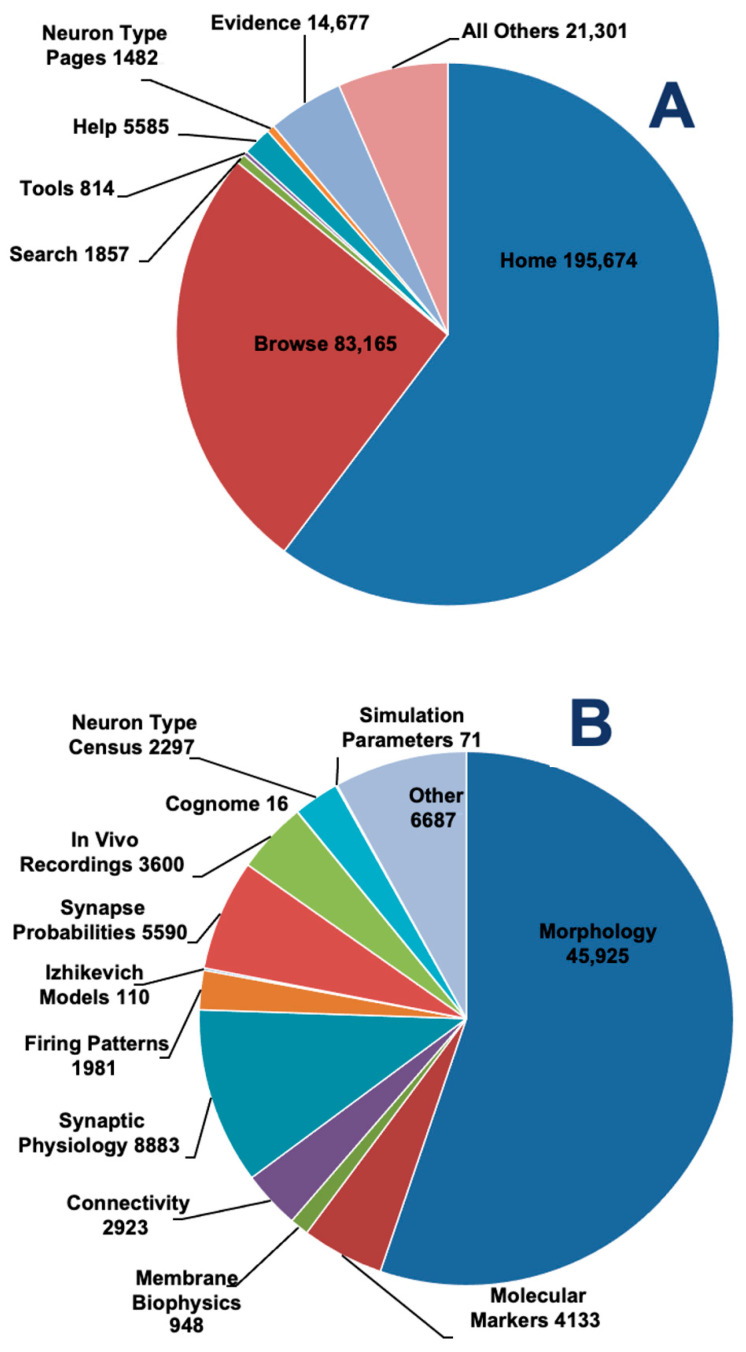
Decomposing Access to the Hippocampome.org web portal. (**A**) User interactions grouped by major navigational functions of the site. (**B**) Page views within the Browse menu, underscoring relative user attention to distinct data dimensions.

**Figure 6 biomedicines-13-02363-f006:**
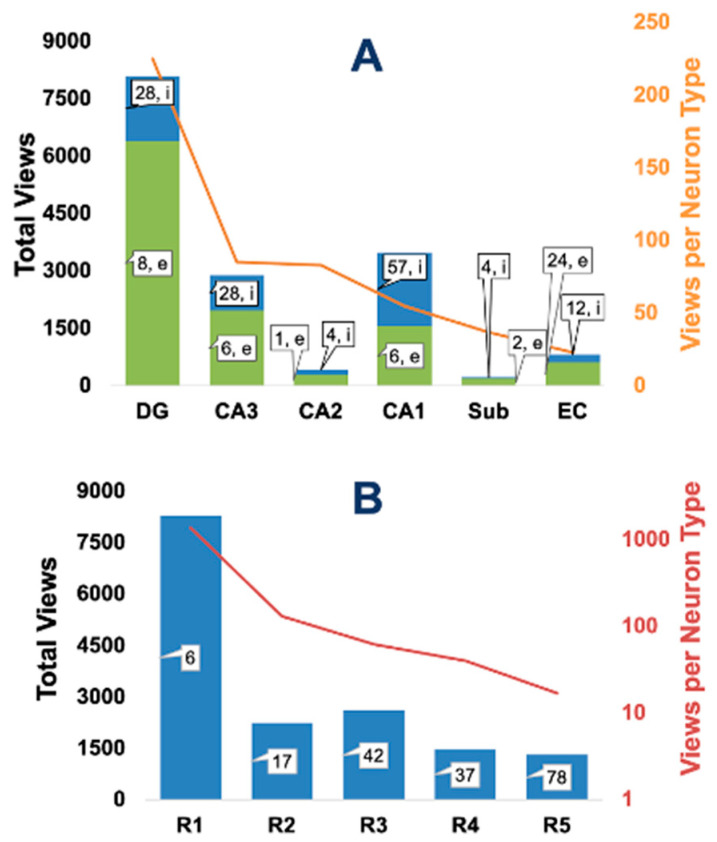
User engagement by hippocampal region and neuron types. (**A**) Page views by region, further stacked into excitatory (green) and inhibitory (blue) neuron types. Numerical labels within or near the bars indicate the count of neuron types in each category. The overlaid orange line (right axis) marks the normalized views per neuron type. (**B**) Page views by Hippocampome.org neuron type rank (R1-5); the red line (right axis) denotes normalized views per neuron type on a log scale.

**Table 1 biomedicines-13-02363-t001:** Summary of Hippocampome.org data domains, property types, and knowledge content.

Property	Property Type	N. Properties	Description	N. PoK	Examples
Morphology	Qual. & Quant.	130	Presence/absence of axon, dendrite & soma in each hippocampal parcel + axonal and dendritic lengths & somatic distances	2927	DG basket cells have axons in the granular layer & dendrites in all DG layers
Molecular Markers	Qual.	118	Expression of enzymes, peptides, channel subunits, etc.	1924	CA1 O/L-M interneurons express somatostatin
Membrane Biophysics	Quant.	10	intrinsic passive and active electrophysiological features	944	DG granule cells V_rest = −75 ± 2 mV
Connectivity	Qual. & Quant.	360	Neuron type potential synaptic circuit, connection prob. & N. contacts	17,546	CA3 pyramidal neurons project to CA1 Pyramidal Neurons via the Schaffer collateral pathway
Synaptic Physiology	Quant. & Model fit	610	Synaptic transmission and short-term plasticity parameters & detailed experimental conditions	499,200	The conductance of CA1 neurogliaform proj. cells onto DG semilunar granule cells in P14 male rats at RT at −60 mV is 1.13 nS
Firing Patterns	Quant.	48	Characteristic spiking response to somatic stimulation	3982	CA1 bistratified interneuron exhibit persistent stuttering
Izhikevich Models	Model fit	9	Computational simulations of neuron spiking and bursting dynamics	2010	The recovery sensitivity (param. b) of subicular CA1-projecting pyramidal cells is 25
In Vivo Recordings	Quant.	12	Real-time electrophysiological data from live rodents	128	CA3 basket cells have an in vivo θ phase of 182 deg.
Neuron Type Census	Error minim.	2	Estimated count of neurons in each type for rat and mouse	284	The rat MEC has ~10,800 L2 spiny stellate cells
Total		1169		528,945	

**Table 2 biomedicines-13-02363-t002:** Most viewed properties and property-specific Pieces of Knowledge (PoK) across data domains.

Property	Most Viewed Property	Property Views	Most Viewed PoK	PoK Views
Morphology	Axon Locations	2440	DG Granule neurons project to Hilus layer	196
Molecular Markers	CB	2616	CB Positive in DG Granule neurons	252
Membrane Biophysics	V_thresh_	14	28 V_thresh_ for CA3 Pyramidal neuron	5
Connectivity	N. Potential Synapses	1006	0.01162 potential synapses from DG Granule cells to DG Hilar Ectopic Granule cells	262
Firing Patterns	Adaptive Spiking	994	DG Granule cells display adaptive spiking	404
In Vivo Recordings	θ phase	1967	124 Theta phase for DG Granule neuron	765

## Data Availability

Data are contained within the article.

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
