# Peer review of "Organization and Community Usage of a Neuron Type Circuitry Knowledge Base of the Hippocampal Formation"

_biomedicines, 2025, doi:10.3390/biomedicines13102363_

Round 1

Reviewer 1 Report

Comments and Suggestions for Authors

In this research article with a title Organization and Community Usage of a Neuron Type Circuitry Knowledge Base of the Hippocampal Formation.” authored by Nadella and his co-authors, the authors have discussed the usage of a web source (https://www.hippocampome.org) based on various factors such as geographical regions, nature of search, utility, and so on. As it is a further and detailed explanation by the same author who has already authored many articles on the same source.

  1. https://pmc.ncbi.nlm.nih.gov/articles/PMC10942544/
  2. https://elifesciences.org/articles/09960
  3. https://link.springer.com/rwe/10.1007/978-1-0716-1006-0_100676
  4. https://journals.plos.org/plosbiology/article?id=10.1371/journal.pbio.3001213

Therefore, the current investigation highlights the different aspects of this source utilized by the scientific community. The purpose of this article seems to be to strengthen its utility and authenticity. Moreover, it is a well-designed and well-drafted manuscript; however, I suggest rethinking the current of this article. The abstract requires revision to make it more understandable, as currently it provides a review on the hippocampus.

Author Response

Comment: "I suggest rethinking the current of this article. The abstract requires revision to make it more understandable, as currently it provides a review on the hippocampus."

Thank you! We have revised the abstract using a structured format to make it more understandable.

Reviewer 2 Report

Comments and Suggestions for Authors

Thanks for the invitation to review this work. This is a well-structured study detailing the development, analytics, and global impact of Hippocampome.org. The integration of Google Analytics to quantify user engagement is innovative and provides actionable insights for future enhancements. The work advances neuroinformatics by bridging experimental data with computational modeling.

  1. Table 1: The "Total" row for  Propertiessums to 1,299, but the text states "over 1,000 distinct properties" (line 209). Clarify this discrepancy.
  2. Address GDPR/ethics compliance for GA tracking (Section 2).
  3. Briefly detail encryption standards (e.g., HTTPS/TLS) for token-based auth (Section 3.7).
  4. Define criteria for neuron-type ranking (e.g., "R5: single publication") earlier in Section 3.5 (currently only in Fig 6B caption).
  5. For Methods: Condense GA4 pipeline details (Section 2) by referencing GitHub repositories more prominently.
  6. Explicitly state how this work transcends prior Hippocampome.org publications (e.g., v1.0–v2.0 updates).
  7. Expand brief mentions of cross-species comparisons (line 399) or LLM interfaces (line 400) into standalone paragraphs.
  8. Typos/Formatting

Line 392: "Emphsis" → "Emphasis".

Line 220: "Total" row in Table 1: "528,945" → Align commas (528,945).

References: Unresolved "[25]" (Google Analytics) and "[21]" (Looker Studio).

  1. Consider including citations of Exploration, 2024, 4, 20230146

Author Response

Table 1: The "Total" row for  Propertiessums to 1,299, but the text states "over 1,000 distinct properties" (line 209). Clarify this discrepancy.

--> Thank you. We have edited the text into "nearly 1300 distinct properties"

Address GDPR/ethics compliance for GA tracking (Section 2).

--> We have added a section in the Methods.

Briefly detail encryption standards (e.g., HTTPS/TLS) for token-based auth (Section 3.7).

--> We have added two sentences in the Methods.

Define criteria for neuron-type ranking (e.g., "R5: single publication") earlier in Section 3.5 (currently only in Fig 6B caption).

--> We moved the definition to before Fig. 6.

For Methods: Condense GA4 pipeline details (Section 2) by referencing GitHub repositories more prominently.

--> We referenced explicitly GitHub twice in this section. We reviewed the GA4 pipeline details and feel that they are necessary to clearly allow others to reproduce our results.

Explicitly state how this work transcends prior Hippocampome.org publications (e.g., v1.0–v2.0 updates).

--> We now do this at the end of the Introduction.

Expand brief mentions of cross-species comparisons (line 399) or LLM interfaces (line 400) into standalone paragraphs.

--> We expanded the mention of cross-species comparison by linking it to the opportunity for technological applications, and LLM interfaces with an additional reference to information extraction from structured databases. 

Typos/Formatting

Line 392: "Emphsis" → "Emphasis".

--> Corrected (thank you!)

Line 220: "Total" row in Table 1: "528,945" → Align commas (528,945).

--> The table was type-setted by the MDPI copyeditor as far as we understand

References: Unresolved "[25]" (Google Analytics) and "[21]" (Looker Studio).

--> Corrected, thank you.

Consider including citations of Exploration, 2024, 4, 20230146

--> Included.

Reviewer 3 Report

Comments and Suggestions for Authors

The manuscript “Organization and community usage of a neuron type circuitry knowledge base of the hippocampal formation” by Nadella is a research article which integrated Google Analytics (GA) into the platform to monitor user behavior, identify high-impact content, and evaluate geographic reach for understanding the diverse neuron types within the hippocampal formation. The authors found that GA data provided valuable page view metrics, revealing usage trends, frequently accessed neuron properties, and the progressive adoption of new functionalities. Such insights directly inform iterative development, particularly in the design of a robust Application Programming Interface (API) to support programmatic access. In general, this article is critical in this field and contains essential contents. I have minor concerns before this manuscript is accepted for publication.

Please explain why the authors examined VGLUT3 as molecular marker expression for CA2 neuron types!

Author Response

"Please explain why the authors examined VGLUT3 as molecular marker expression for CA2 neuron types!"

Thank you. This was just chosen as an example of molecular biomarker, but we have now added a brief explanation where this chemical is mentioned.